# The Effects of Fermentation of Low or High Tannin Fava Bean-Based Diets on Glucose Response, Cardiovascular Function, and Fecal Bile Acid Excretion during a 28-Day Feeding Period in Dogs: Comparison with Commercial Diets with Normal vs. High Protein

**DOI:** 10.3390/metabo11120878

**Published:** 2021-12-16

**Authors:** Luciana G. Reis, Tressa Morris, Chloe Quilliam, Lucas A. Rodrigues, Matthew E. Loewen, Lynn P. Weber

**Affiliations:** 1Department of Veterinary Biomedical Sciences, Western College of Veterinary Medicine, University of Saskatchewan, Saskatoon, SK S7N 5B4, Canada; lg.reis@usask.ca (L.G.R.); tressa.morris@usask.ca (T.M.); cjq897@mail.usask.ca (C.Q.); matthew.loewen@usask.ca (M.E.L.); 2Department of Animal and Poultry Science, College of Agriculture and Bioresources, University of Saskatchewan, Saskatoon, SK S7N 5A8, Canada; lucas.rodrigues@usask.ca; 3Prairie Swine Center, Inc., Saskatoon, SK S7H 5N9, Canada

**Keywords:** fava bean (*Vicia faba*), fermentation, cardiovascular function, blood chemistry, glucose tolerance, domestic dog (*Canis familiaris*)

## Abstract

We have shown that feeding dogs fava bean (FB)-based diets for 7 days is safe and FB flour fermentation with *Candida utilis* has the potential to decrease FB anti-nutritional factors. In the present study, the effects of 28-day feeding of 4 different FB-based test dog foods containing moderate protein (~27% dry matter (DM)) were compared with two commercial diets with normal protein (NP, grain-containing, ~31% DM protein) or high protein (HP, grain-free, ~41% DM protein). Health parameters were investigated in beagles fed the NP or HP diets or using a randomized, crossover, 2 × 2 Latin square design of the FB diets: unfermented high-tannin (UF-HT), fermented high-tannin (FM-HT), unfermented low-tannin (UF-LT), and fermented low-tannin (FM-LT). The results showed that fermentation increased glucose tolerance, increased red blood cell numbers and increased systolic blood pressure, but decreased flow-mediated vasodilation. Taken together, the overall effect of fermentation appears to be beneficial and improved FB nutritional value. Most interesting, even though the HP diet was grain-free, the diet did contain added taurine, and no adverse effects on cardiac function were observed, while glucose tolerance was impaired compared to NP-fed dogs. In summary, this study did not find evidence of adverse cardiac effects of pulses in ‘grain-free’ diets, at least not in the relatively resistant beagle breed over a 28-day period. More importantly, fermentation with *C. utilis* shows promise to enhance health benefits of pulses such as FB in dog food.

## 1. Introduction

Fava bean (*Vicia faba* L.) is a proteinaceous ingredient with potential for partially replacing animal protein sources in human diets and has been associated with health improvements triggered by its functional properties [1]. For companion animals, despite the increased interest in incorporating fava beans in their diets, there are still concerns about potential toxicity from anti-nutritional factors [2], which prevent fava bean’s approval as a feed ingredient for pet food by the American Association of Feed Control Officials (AAFCO). Moreover, complicating the approval of fava beans is a concern surrounding all pulses used in pet food. Specifically, the US Food and Drug Administration [3] reported that pulses were linked to dilated cardiomyopathy (DCM) occurrence in dogs fed grain-free diets (i.e., food formulated with potatoes and pulse ingredients instead of grains). While the underlying cause and verification of this link remain unsolved, proof of safety for any pulses, particularly ones new to pet food, such as fava beans, is needed.

In dogs, DCM is one of the most prevalent, genetically-associated cardiac diseases and is most common in large or giant breeds [4]. One proposed mechanism by which nutrition may link to DCM is through taurine insufficiency [5,6]. Taurine is an important component of calcium reabsorption pathways within the sarcoplasmic reticulum and increases the sensitivity of the cardiac myofilaments to calcium [7]. Thus, it is reasonable to infer that under suboptimal intake of taurine or its precursors (e.g., methionine, cysteine), cardiac contraction could be compromised which in turn increases risk of developing DCM [8]. Since grain-free and pulse-containing diets tend to be high in fiber, dogs fed these diets may have decreased protein digestibility due either to decreased bioavailability of sulfur-containing amino acids or to higher fecal excretion of taurine via taurocholate [9,10,11,12]. Plant proteins are naturally low in taurine precursor amino acids and completely lack taurine, compounding the taurine insufficiency issue.

There are several anti-nutritional factors in fava beans including condensed tannins, trypsin inhibitor activity, lectins, and pyrimidine glucosides that impair nutrient digestibility [2,13]. Importantly, consumption of uncooked fava beans has been associated with sudden red blood cell death and acute anemia in humans with favism, a genetic mutation of glucose-6-phosphate dehydrogenase (G6PD) that increases sensitivity to pyrimidine glucosides (vicine and convicine) toxicity. Fava bean fermented with *Lactobacillus plantarum* showed reduced vicine and convicine concentrations, with greater availability of free amino acids and protein digestibility [14,15,16,17]. Furthermore, fermentation using the yeast *Candida utilis* (*C. utilis*) may bring the dual benefit of reducing anti-nutritional factors and increasing taurine content [18].

We recently reported that fava bean–based diets did not cause hemolytic anemia and did not alter glucose response in dogs after a 7-day feeding period [19]. Interestingly, fermentation with *C. utilis* reduced the concentration of anti-nutritional factors and improved nutrient digestibility and red blood cell counts. Furthermore, fava bean-based diets were contrasted with commercial diets with normal or high protein, with the latter showing detrimental effects on blood chemistry, but improved cardiovascular functioning of dogs. Since the negative effects observed in our previous study could impact cardiovascular health to a greater extent if diets were fed longer than 7 days, studies using longer feeding periods are needed to assess safety.

Therefore, the objective of this study was to determine the effect of longer-term (28-day) feeding of beagles with unfermented or fermented fava bean-based diets on glucose response, body weight, cardiovascular function, blood parameters, and fecal bile acid content when contrasted to commercial diets with normal vs. high protein. It was hypothesized that pulse-based diets would impair cardiovascular health and increase blood markers of cardiac damage (cardiac troponins, N-terminal pro-brain natriuretic peptide or NT-proBNP) due to the low taurine, cysteine, or methionine levels and high fiber content. Moreover, fermentation of fava bean flour with *C. utilis* would enhance diet quality and, consequently, health of dogs.

## 2. Results

### 2.1. Body Weight, Meal Portion, and Body Condition Score

Body weight (BW), body condition score (BCS), and meal portion data are presented in Table 1. Body condition score was assessed using a 9-point scale [20]. No significant effect (*p* > 0.05) of dietary protein content (NP vs. HP) was observed on BW, BCS or meal portion after 28 days of feeding in beagles. Likewise, within fava bean–based diets, there was no effect of either FM, FB, or the interaction between FM and FB on BW, BCS, or meal portion (*p* > 0.05).

### 2.2. Glucose Response

Blood glucose responses to the oral glucose tolerance test are shown in Figure 1. Baseline (0 min) blood glucose levels were not significantly different between NP and HP (*p* > 0.05) while HP-fed dogs showed greater glucose levels compared to NP at 30 min after glucose feeding (*p* < 0.05). Also, there was no effect of fava bean variety or fermentation at 0 min, while dogs fed FM-HT had lower glucose levels at 30 min after glucose challenge compared to UF-HT, with UF-LT and FM-LT being intermediate (*p* < 0.05).

### 2.3. Red and White Blood Cell Count

Red (RBC) and white blood cell (WBC) counts of beagles after feeding them each diet for 28 days are shown in Figure 2. There was no effect of dietary protein content on WBC or RBC levels (*p* > 0.05). Within fava bean–based diets, FM-LT dogs had greater RBC compared to UF-LT, with UF-HT and FM-HT being intermediate (*p* < 0.05). White blood cell count decreased (*p* < 0.10) in dogs fed FM compared to UF diets regardless of fava bean variety.

### 2.4. Blood Parameters of Hepatic Function

Blood parameters indicative of hepatic function in beagles after feeding them test diets for 28 days are shown in Table 2. Within commercial diets, indirect bilirubin was lower while urea was greater in HP-fed dogs compared to NP (*p* < 0.05). Moreover, glutamate dehydrogenase decreased in NP compared to HP dogs (*p* < 0.10). Within fava bean-based diets, indirect bilirubin increased in HT compared to LT dogs regardless of fermentation (*p* < 0.10). Furthermore, gamma-glutamyl transferase decreased while alanine aminotransferase increased in FM compared to UF dogs regardless of fava bean variety (*p* < 0.10). There was an interaction between fava bean variety and fermentation where direct bilirubin was decreased in FM-HT compared to UF-HT dogs, with UF-LT and FM-LT being intermediate (*p* < 0.05). No effects of dietary protein content, fava bean variety or fermentation were observed on cholesterol, total bilirubin, alkaline phosphatase, creatine kinase, total protein, albumin, globulin, albumin:globulin or creatinine levels (*p* > 0.05).

### 2.5. Blood Electrolytes and Enzymes

Blood electrolytes from beagles after feeding them each test diet for 28 days are shown in Table 3. Within commercial diets, Na, anion gap, and Mg were increased in HP dogs compared to NP (*p* < 0.05). Moreover, Cl decreased (*p* < 0.05) while P increased (*p* < 0.10) in HP-fed compared to NP-fed dogs. Within fava bean-based diets, FM-fed dogs have a greater anion gap compared to UF-fed dogs regardless of fava bean variety (*p* < 0.10). Moreover, FM increased lipase levels compared to UF regardless of fava bean variety (Figure 3, *p* < 0.05). Finally, there was an interaction between fava bean variety and fermentation where UF-LT dogs had greater P than UF-HT dogs, with FM-LT and FM-HT being intermediate (*p* < 0.05). No effects of dietary protein content, fava bean variety or fermentation were observed on K, Na:K, HCO_3_^−^ or Ca levels (*p* > 0.05).

### 2.6. Cardiovascular Function

Cardiovascular function parameters of beagles after 28 days of feeding them each test diet are shown in Table 4. Feeding HP diets decreased left ventricular end systolic diameter, increased stroke volume, cardiac output (Simpson’s rule measurements only), ejection fraction, and maximum velocity (A wave), while decreasing diastolic blood pressure compared to NP (*p* < 0.05). Moreover, dogs fed HP have greater left ventricular end-diastolic volume compared to NP dogs (*p* < 0.01 for Simpson’s rule). Within fava bean-based diets, HT-fed dogs had greater left ventricle diastolic wall thickness and lower maximum velocity (A wave) compared to LT, regardless of fermentation (*p* < 0.05). Additionally, feeding FM diets decreased ejection fraction (Simpson’s rule measurements only), increased diastolic blood pressure, and decreased FMD (flow mediated dilation) compared to UF diets, regardless of fava bean variety (*p* < 0.05). Moreover, left ventricular end-systolic volume increased in FM compared to UF regardless of fava bean variety (*p* < 0.10) measured with M-mode, but not Simpson’s rule. Left ventricle systolic wall thickness was greater in UF-HT dogs compared to the other groups (*p* < 0.05). Also, systolic blood pressure was greater in FM-LT compared to UF-LT, with UF-HT and FM-HT being intermediate. Finally, left ventricular end-diastolic diameter, heart rate, maximum velocity (E wave), stroke volume (M-mode), cardiac output (M-mode), and E/A ratio were not altered by dietary protein content, fava bean variety or fermentation.

Regarding plasma levels of cardiac troponin I and NT-proBNP, no effects of dietary protein level, fava bean variety or fermentation were observed on those cardiac biomarkers.

### 2.7. Digestibility

Apparent total tract digestibility in beagles fed fava bean–based diets are shown in Table 5. Dogs fed FM diets have greater non-fiber carbohydrate digestibility compared to UF diets regardless of fava bean variety (*p* < 0.10). Crude protein digestibility was greater in FM-LT compared to FM-HT, with UF-LT and UF-HT being intermediate (*p* < 0.05). Moreover, gross energy digestibility was increased in FM-LT-fed dogs compared to FM-HT, with UF-LT and UF-HT being intermediate (*p* < 0.05). No effects of dietary protein content, fava bean variety or fermentation were observed on fat, methionine, cysteine, or taurine digestibility.

### 2.8. Plasma Amino Acid Levels

Plasma amino acid concentrations in beagles after 28 days of feeding them each test diet are shown in Table 6. Dogs fed HP diets have increased cystine concentration compared to NP-fed dogs (*p* < 0.10). Fermentation decreased cystine concentration compared to UF regardless of fava bean variety (*p* < 0.05). Beagles presented no significant differences in taurine and methionine plasma concentrations regardless of diet protein content, fava bean variety or fermentation.

### 2.9. Fecal Bile Acid Content

Fecal bile acid levels are shown in Figure 4. No effect of dietary protein content was observed (*p* > 0.05). Within fava bean-based diets, UF-LT and FM-HT dogs showed greater fecal bile acid levels compared to FM-LT, with UF-HT being intermediate (*p* < 0.05).

## 3. Discussion

We have previously reported that 7-day feeding of fava bean diets with moderate protein levels formulated to just meet the AAFCO [21] requirements for methionine (0.33%) or methionine + cystine (0.65%) did not produce clinical or physiological signs associated with cardiovascular impairment in adult beagle dogs [19]. The most important finding in this current study was that dogs fed fava bean-based diets (30% inclusion) for 28 days again did not present impaired overall or cardiovascular health, with the plasma amino acids at safe and satisfactory levels. Furthermore, the high protein, grain-free commercial diet tested did not cause cardiovascular dysfunction when fed for the same 28-day period to dogs, but instead showed impaired glucose tolerance.

### 3.1. Toxicity from Fava Beans, Digestibility, Glucose Tolerance, and Anti-Nutritional Factors

In the present study, fermentation decreased the vicine and convicine content in fava bean flours and did not influence plasma taurine content or amino acid digestibility after 28-day feeding, which is in agreement with our previous findings in a short-term feeding period [19]. It has been recently shown that fermentation with *L. plantarum* effectively degraded the pyrimidine glycosides in fava bean flour within 48 h, which attenuated the toxicity of the fermented diets [22]. Conversely, plasma cystine was decreased in dogs fed fermented diets, and protein digestibility was greater in dogs fed fermented low tannin diets compared to fermented high tannin diets. This is contrary to our previous findings where protein digestibility was not influenced by fermentation, while plasma cysteine was increased in dogs fed fermented diets. A decreased plasma cystine content with the same taurine content in dogs fed fermented diets may indicate prioritization of endogenous cystine utilization for synthesis and normalization of plasma taurine concentration. Cystine is the predominant form of cysteine found in plasma and is a more stable, oxidized dimer of two cysteine molecules [23]. However, since we could not measure plasma cysteine due to chemical instability and technical issues, we cannot say with certainty whether total cysteine + cystine would have been changed in the same way as cystine alone. A greater protein digestibility in fermented low tannin compared to fermented high tannin diets may indicate that when feeding pulse-based diets for a longer period, the positive effects of fermentation may be dependent on tannin levels of fava beans prior to fermentation. This is an important finding of the present study since fermentation of proteinaceous ingredients may benefit host overall health due to improvements in essential amino acids and bioactivity through digestion by the micro-organism [24]. Furthermore, fermentation increased dog plasma lipase in the current study, which was accompanied by a decreased fecal bile acid content in dogs fed fermented low tannin diets. These findings are somewhat non-specific and may not be linked, but could indicate an improved bile acid absorption in dogs fed fermented diets. Based on this effect, fermented diets have the potential to improve stool quality, since an excess of bile acids being spilled into the colon may stimulate electrolyte and water secretion, resulting in loose feces [25]. Stool quality was qualitatively observed to be unchanged by diet in the current study, but at no time was feces dry or hard, agreeing with this hypothesis. This is also consistent with historical data in rats, where the fermentability of fibrous ingredients was directly associated with a decrease in bile acid excretion [26]. Finally, fermentation increased RBC content in LT-fed dogs only. Young dogs generally experience low RBC content due to the RBC lifespan which is shorter and with lower hemoglobin content for young RBCs compared with aging RBCs [27]. An increased RBC content in dogs fed diets with fermented low tannin fava bean (Snowdrop variety), but not high tannin (Florent variety), indicates that fermentability depends on fava bean variety. This corroborates previous findings from our group where fermentation provided a higher production of red blood cells [19], although increased RBC lifespan could also explain this finding. Other studies report fermentation also increased short-chain fatty acids that can be linked to decreased insulin and higher high-density lipoprotein level [28].

We used two fava bean varieties in the present study, which are genotypes commonly grown in Canada [29], and generally classified as low-tannin (Snowdrop) [30] and high-tannin (Florent) [29]. However, the proximate analysis revealed high content of vicine and convicine in both fava bean varieties. High tannins could dramatically decrease nutrient bioavailability in the gastrointestinal tract [31,32], particularly during an extended feeding period. We are unable to make inferences about bioavailability in the present study; however, fermentation was observed to increase protein digestibility in low tannin compared to high tannin fava bean varieties and increased gross energy digestibility in low tannin only. These findings contrast with our short-term study where there was no effect of fava bean variety on digestibility values of any nutrient measured [19]. This indicates that the negative effects of anti-nutritional factors may be directly associated with feeding period length. Moreover, the ability of fermentation to improve protein digestibility appears to be dependent on fava bean variety. One remarkable finding of the present study was a decreased glucose response at 30 min after glucose feeding in dogs fed fermented high tannin fava bean-based diets compared to unfermented despite the lack of effect on fasting glucose. This suggests that long-term, fava bean varieties differentially influence glucose tolerance and/or that fermentation effects on glucose tolerance are only observed during a longer feeding period [19]. It should be highlighted that the four fava bean-based diets were fed sequentially in a crossover design for an overall period of four months, with no change in glucose tolerance from the previous NP period (first month). Furthermore, RBC content was within expected ranges in dogs fed fava bean-based diets compared to NP diets, which indicates that, even during a long-term feeding period, there was no anemia in the dogs and therefore favism was not a concern in the present study. Our long-term feeding trial provides additional evidence that fava beans should be further explored as an ingredient for dog food. However, a 6-month feeding study is required by AAFCO to approve a given ingredient to be used for pet food and this remains to be done.

### 3.2. High Dietary Protein Diet on Dog Health

In agreement with our recent study assessing a short-term feeding period [19], we confirmed that high dietary protein can negatively affect the overall health of dogs, albeit to a lesser extent than previously observed. In the present study, urea was increased in HP compared to NP dogs. Higher urea signals kidney overload which could lead to increased proteinuria and potentially impairment of the overall health of dogs [33]. Furthermore, indirect bilirubin content was decreased in HP-fed dogs compared with NP, which is in line with a previous study [34], that reported higher bilirubin levels in young pigs fed a protein restricted diet compared with a control diet. In our short-term feeding trial [19], we observed potential positive effects of feeding HP diets to dogs including decreased ALP and ALT compared with levels of dogs fed NP, which were not confirmed in the present study. Increased ALT content in dogs may be an indicator of hepatocyte membrane damage and necrosis, while increased ALP content may indicate biliary stasis [35,36], which could be interpreted as poor hepatic function [37]. Since these observations were not replicated in a long-term feeding period and all values in this study were within clinical norms, the trends previously reported were likely random variation and are likely clinically insignificant.

### 3.3. Cardiac Function, Cystine, Methionine, and Taurine

The main objective of the present study was to feed pulse-based, high fiber diets to dogs aiming to investigate if the lack of effect on cardiac or vascular function observed after 7 days of feeding [19] would be sustained after 28 days of feeding. Specifically, changes in plasma taurine, cystine, or methionine levels, and/or alterations in cardiac contractility or enlargement of the heart consistent with DCM are of interest. Feeding high tannin fava bean-based diets increased left ventricle diastolic wall thickness and decreased the maximum velocity of left ventricular filling during atrial contraction (A wave) compared to low tannin. This could indicate an elevated left ventricular mass [38] which in conjunction with a decreased maximum velocity could be an initial signal of cardiac dysfunction. However, it should be noted that while statistically significant changes were observed in the current study, all values remained within clinical norms, indicating that the dogs remained healthy. Supporting the conclusion of cardiac health is the fact that neither NT-proBNP nor cardiac troponin levels changed with diet in the current study. These proteins, when present in the blood or plasma, are indicators of heart failure and cardiac damage, respectively [39,40]. Thus, our data do not support the hypothesis that fava bean-based dog foods cause cardiac impairment or DCM.

Surprisingly, diastolic blood pressure was increased, and flow mediated dilation was decreased in dogs fed fermented diets compared to unfermented regardless of fava bean variety, both of which would be considered adverse changes in vascular health. Flow mediated dilation, which is the vasodilatation of an artery following an increase in luminal blood flow and internal-wall shear stress, has been widely used as a surrogate, non-invasive marker of vascular health. The measurement of FMD is positively associated with endothelial nitric oxide synthase expression, and nitric oxide bioactivity [41,42]. As stated previously, feeding fermented diets produced a decreased FMD accompanied by an increase diastolic blood pressure. These effects are consistent with nitric oxide quenching due to increased oxidative stress which can be counteracted through dietary consumption of many plant polyphenols that act as antioxidants [43,44,45]. It appears that oxidative stress promotes cellular damage resulting in endothelial dysfunction and a gradual worsening of vascular health [46]. Fermentation may have played a role in reducing the beneficial antioxidant effects while at the same time reducing the anti-nutritional factors in fava bean such as vicine/convicine. Considering that the animals were healthy, and no clinical or physiological effects were observed, losing the toxic compounds provides greater benefit than negative from a slight loss in antioxidant capacity. On the other hand, the present study also provides indication of no adverse effects of feeding HP diets, similar to what was observed at the short-term [19]. Left ventricular end systolic diameter and diastolic blood pressure were decreased while ejection fraction was increased in HP compared to NP dogs. In dogs, increased left ventricular end systolic diameter has been associated with dilated cardiomyopathy [47] and decreased diastolic blood pressure is prognostic of poor outcomes in human heart failure [48]. Thus, given the fact that HP diets did the opposite and decreased end systolic diameter while increasing cardiac function (increased stroke volume and cardiac output), the decreased diastolic blood pressure should not be considered adverse. Conversely, decreased ejection fraction is a potent predictor of mortality and post-operative left ventricular dysfunction in human patients with chronic cardiovascular dysfunction [49,50]. In dogs, Adin et al. [51] showed that larger left ventricular end diastolic diameter, increased left ventricular end systolic diameter and lower sphericity index (left atrial to aortic ratio) indicate advanced myocardial dysfunction and remodeling in dogs. Moreover, left ventricular dilation and depressed systolic function, that can be assessed as reduced ejection fraction and decreased stroke volume and/or cardiac output, are abnormalities required to be present in order to diagnose dilated cardiomyopathy cases [52]. However, of these diagnostic changes, only increased end diastolic volume and ventricular diameter were found with the dogs fed the HP grain-free diet in the present study and in fact, ejection fractions increased. Increased preload is likely to have led to the observed increase in end diastolic volume and diameter [47], but the significance of this is unclear and no other indicators of cardiac function support any adverse effects with the grain-free HP diet compared to grain-containing NP diet. However, it should be noted that the HP grain-free diet contained added taurine which may have ameliorated any potential adverse effects.

The laboratory-formulated diets used in the present study had no significant differences in crude fiber content when compared to the commercial diets. Of note, cysteine content was decreased in fava bean-based diets compared to commercial diets. Moreover, methionine and taurine contents were higher in HP compared to either the NP diets or all fava bean–based diets. An increased dietary taurine content or taurine supplementation is routinely used by veterinarians and nutritionists to improve cardiovascular functioning in dogs with known or suspected taurine deficiency. Interestingly, our results from the current study’s longer-term feeding period are in agreement with that reported for a short-term feeding period e.g., 7 days [19], where plasma levels of taurine were not altered by diet and remained within the desired reference range in our beagles. Future studies should investigate whether more susceptible dog breeds (e.g., Doberman Pinscher, Great Dane, Boxer, and Cocker Spaniel) would show altered cardiovascular response when under the same dietary regimen.

Analyzed dietary methionine content was at AAFCO guideline levels or slightly suboptimal in NP and all fava bean-based diets. Sulfur-containing amino acids have a very dynamic metabolism, where dogs are able to synthesize taurine from cysteine or methionine [53]. When plant-based protein sources such as pulses are fed, these amino acids are generally limiting due to lower levels in ingredients and/or high fiber content which decreases protein digestibility and increases fecal bile acid excretion [54]. The mechanism explaining effects of fecal bile acid excretion relies on high fiber content depleting taurine and decreasing protein digestion, which will further require cysteine and methionine to replace it. Contrary to our short-term study [19], fermented fava bean-based diets (both varieties) led to reduced plasma cystine levels after 28 days feeding each diet, although the cysteine measurements were the least reliable in this study due to instability. Despite this change in plasma cystine, taurine and methionine plasma levels were unaltered and levels remained above the reference range [55], possibly suggesting the cysteine results were spurious. Interestingly, fecal bile acid excretion was decreased in fermented, low tannin, fava bean-based diets. These findings combined suggest that reduced plasma cystine content could be associated with increased utilization for production of taurine in dogs fed fermented diets and that fermentation was able to attenuate fecal bile acid malabsorption (dependent on fava bean variety). Also in contrast to our short-term findings [19], there was no difference among commercial diets on cysteine, methionine, or taurine levels in the present study, which suggests that the alterations of blood levels of these amino acids observed after 7 days of feeding were not sustained in the long-term. Physiological compensation through adjustments in sulfur-containing amino acids metabolism may be responsible for lack of effect after longer term feeding.

## 4. Materials and Methods

The animal study was reviewed and approved by University of Saskatchewan Animal Research Ethics Board (Protocol #20190055).

### 4.1. Fava Bean Ingredients and Fermentation Protocol

Dehulled LT (Snowdrop) and HT (Florent) fava bean varieties, genotypes cultivated in Saskatchewan [29], were ground into flour via a 400-μm screen. The methodology to ferment each type of fava bean was adapted from a process previously used in our lab in pea flour [56]. Concisely, *C. utilis* (ATCC 9950) was preserved in sterile 80% (*v*/*v*) glycerol solution at −80 °C, before being reactivated in YGC agar plates when required (Yeast Extract Glucose Chloramphenicol Agar; catalogue number 95765; Sigma Aldrich, St. Louis, MO, USA). Incubation of seeded plates occurred at 30 °C for 72 h. A flame-sterilized platinum needle was used to transfer two loops of colonies to a 250-mL sterile conical flask containing 100 mL of YPD liquid medium (Yeast Peptone Dextrose—A1374501; ThermoFisher, Waltham, MA, USA). For 12 to 15 h the flask was incubated in a horizontal shaker incubator (30 °C) at 120 rpm. The next step was to transfer 10 mL of the cultured yeast mass into a 500 mL sterile conical flask containing 250 mL of YPD liquid medium. Following, the medium with the yeast was incubated on a horizontal shaker (30 °C) at 120 rpm for an additional 12 to 15 h. A total amount of 20 kg batches of each fava bean genotype were blended with the yeast broth, ammonia, and sterile water to form a soft dough, then the fermentation occurred in an adapted cement mixer with the temperature maintained at 30 °C. The fermentation slurry was mixed for 3 min every hour for 72 h. To prove yeast viability and count throughout the process, samples were collected every 24 h, and serial dilutions incubations at 30 °C for 72 h of mixture plated on sterile agar plates were performed. The fermented fava bean flour was then dried in an oven at 60 °C for 48 h at WCVM prior to transport to the University of Saskatchewan Canadian Feed Research Centre (North Battleford, SK, Canada) for the grinding processing of the fermented flours; after that, all test diets (both fermented and unfermented fava bean flours) were mixed, extruded, and vacuum-coated with fat to create the final dry kibble format of the diets to be consumed in the feeding trials.

### 4.2. Animals and Diets

The University of Saskatchewan Animal Care Committee (Animal Utilization Protocol #20190055) approved the animal use and procedures which adhere to the Canadian Council on Animal Care. In total, eight neutered beagles, four females and four males, at a mean age of 3.3 ± 0.9 years and, at ideal body weight (9.4 ± 2.4 kg), were acquired from King Fisher International (Toronto, ON, Canada) or Marshall Bioresources (North York, NY, USA) and accommodated at the Western College of Veterinary Medicine (Saskatoon, SK, Canada). The beagles were sheltered individually in 1.1 × 2.7 m kennels with access to the outdoors through a kennel door at night and during feeding but kept in a group kennel area during the day. Every day for at least 1 h, volunteers walked or socialized with the dogs. To reduce the stress, all beagles were familiarized to procedures with rewards prior to experiments starting, so no anesthetics or sedatives were used in this study.

Six diets, in total, were tested. Low- or high-tannin fava bean flours, either fermented or unfermented, were used at 30% inclusion, with diet formulations indicated in Appendix A. The concentration of fava bean was chosen based on the knowledge that more than 30% could cause gastrointestinal issues due to the amount of antinutritional factors present in legumes seeds in general. However, there was a need to explore high amounts to investigate the possibility of issues like those reported in humans so high levels of inclusion would give us the possibility to detect any adverse effect from fava bean in beagles. The four fava bean test diets were unfermented low-tannin [UF-LT; 27% crude protein (CP)], fermented low-tannin (FM-LT; 28% CP), unfermented high-tannin (UF-HT; 27% CP), and fermented high-tannin (FMHT; 28% CP). As a non-digestible marker, insoluble ash (Celite) was added at 1%. To achieve nutritionally balanced diets before extrusion under equal process conditions, diets were formulated in agreement with the nutrient guidelines for adult dog maintenance set by AAFCO [21]. The two extra diets tested were commercial diets containing normal-protein (NP; 31% CP) or high protein content (HP; 41% CP). Ingredient lists for the commercial diets are shown in Appendix A. The proximate analysis of all six diets were performed (Central Testing, Winnipeg, MB, Canada) after being randomly subsampled (Appendix A). The analyzed levels of crude fiber, non-fiber carbohydrates, metabolizable energy, vicine, and convicine of UF-LT, UF-HT, FM-LT, and FM-HT diets are 1.1, 0.5, 0.5, and 0.5%; 60.7, 62.1, 61.6, and 61.8%; 3.7, 3.8, 3.8, and 3.8 kcal/g; 1.8, 1.8, 0.4, and 0.6 mg/g; and 0.5, 0.6, 0.1, and 0.2 mg/g, respectively. For crude fiber, non-fiber carbohydrates, and metabolizable energy of NP and HP diets, the analyzed contents were 0.8 and 1.1%, 53.0 and 29.8%, and 4.0 and 4.1 kcal/g, respectively. The feeding period of each diet occurred for 28 days, with the NP or HP diets being used on the first and sixth months, respectively. From months 2 to 5, using a randomized, crossover, 2 × 2 Latin square design, the UF-LT, FM-LT, UF-HT, and FM-HT diets were fed. The two commercial diets were not incorporated into the crossover design as the fava bean–based diets were formulated to include the same nutrient profile to enable a satisfactory comparison. On the other hand, NP and HP diets have different components at unidentified inclusion amounts (specific formulation not available on label), making direct comparisons problematic. All beagles were weighed at the start of the experiment and after each feeding period. Standard equations were used to determine the energy requirements for individual dog maintenance [maintenance energy (ME in kcal)] = [(70 × BW0.7) ×1.6], so the quantity of diet assigned per dog per day is isocaloric with the daily requirement for that dog; daily amounts were apportioned, and dogs were fed twice daily (at 08:30 and 16:30 h). Before the following meal, bowls were removed, and any food left uneaten was weighed and recorded. No palatability issues were observed [56] as all dogs usually ate all food portioned in each meal within the period of 5–10 min.

### 4.3. Digestibility Protocol

Feces were collected for 2 subsequent days after a 26-day feeding period on each diet (total of 28 days on each diet). Feces were stored in containers and labeled, and frozen at −20 °C until analysis. Feces samples were thawed, homogenized, and pooled by dog (i.e., two samples from each dog pooled per dietary treatment). Prior to laboratory testing, feces were dried in a forced air oven at 55 °C for 72 h and ground in a cutting mill with a 1 mm sieve. Dry matter by oven-drying the sample, non-fiber carbohydrates, crude protein applying the Kjeldahl method, and acid-hydrolyzed fat were analyzed (Central Testing, Winnipeg, MB, Canada) from feces and diets, according to AOAC standards [57]. A bomb calorimeter was used to determine the content of gross energy (GE) of diets. To calculate the apparent digestibility coefficients of dry matter, crude protein, non-fiber carbohydrates, fat, gross energy, methionine, cysteine, and taurine for each fava bean–based diet using Celite as a non-digestible marker [58], the following equation was used:Digestibility (%) = 100 − [100 × (Mfeed × Cfeces/Mfeces × Cfeed)]
where Cfeed and Cfeces indicate amount of components of interest in feed and feces, respectively; Mfeed and Mfeces indicate concentration of index compound in feed and feces, respectively.

### 4.4. Oral Glucose Response Test and Blood Analysis

A 10% glucose solution (1 g/kg BW glucose) was fed to each dog at the same time each day, after 28 days of feeding each diet and an overnight 8 h fasting, by placing a syringe with the solution in the back of the mouth, for the conduction of an oral glucose tolerance test. Previous to the glucose feeding, the fasted dogs were aseptically catheterized in the cephalic vein using a peripheral intravenous catheter equipped with an extension tube. Before feeding glucose (time 0) and at 30 min after feeding, as 30 min was previously observed as the glucose peak concentration [19], blood samples (∼0.2 mL) were collected. Blood glucose was measured using a glucometer (OneTouch Ultra 2; LifeScan, Johnson & Johnson, New Brunswick, NJ, Canada) with at least two readings for each time or until two consistent readings were achieved. To guarantee no contamination of blood with anticoagulant solution, the extension tube was loaded with blood and the initial blood was discarded prior to the glucometer reading. Afterwards, the catheter was flushed with a sterile citrate solution to prevent any clotting right after each blood sample was collected. Extra fasting blood samples (8 mL) were obtained using collection tubes with and without EDTA.

Complete blood cell count [red (RBC) and white (WBC) blood cell counts] and chemistry panel [cholesterol; total (TB), direct (DB), and indirect bilirubin (IB); alkaline phosphatase (ALP); alanine aminotransferase (ALT); creatine kinase (CK); gammaglutamyltransferase (GGP); glutamate dehydrogenase (GLDH); total protein, albumin (A), globulin (G), and A:G] were analyzed at Prairie Diagnostic Services (Saskatoon, SK, Canada). Also, 3 mL subsamples of EDTA-tube blood from fasted animals were centrifuged at 2000 rpm for 10 min for plasma collection. Plasma samples were maintained at −80 °C until further analyses of methionine, cystine, and taurine content (UC Davis Amino Acid Lab, Davis, CA, USA). Unlike our previous study [19], the cysteine level was not analyzed by UC Davis Amino Acid Lab due to high instability of the parent cysteine compound. Plasma amino acid concentrations were analyzed in an automated amino acid analyzer via cation exchange high-pressure liquid chromatography separation and ninhydrin-reactive colorimetric detection [59,60,61].

### 4.5. Cardiac Function, Blood Pressure, and Vascular Health

Dogs were examined for cardiovascular health after 28 days of feeding them each diet. All ultrasound parameters were executed and analyzed by a single sonographer who was a DVM with specific training in ultrasound technique and >200 h echocardiography experience; repeatable measurements in the same beagles were determined prior to the start of the current study. Blood pressure was obtained using a high-definition canine/feline oscillometer (VET HDO High Definition Oscillometer, Babenhausen, Germany), before ultrasound examination. An average of two readings with good agreement was utilized to establish diastolic and systolic pressures. To indicate vascular health endpoints of flow-mediated dilation were used which included brachial artery diameter during baseline, during inflation of a blood pressure cuff placed distal to the brachial artery, and at the time of peak dilation (30 s) after cuff release, as established before by our research group in dogs [19,62,63]. Echocardiography M-mode endpoints used to assess cardiac function included heart rate (HR), stroke volume (SV), and cardiac output (CO) [64,65]. Additionally, left ventricular end-diastolic volume (EDV), left ventricular end-systolic volume (ESV), ejection fraction (EF), left ventricular diastolic free wall thickness (DWT), left ventricle systolic free wall thickness (SWT), systolic blood pressure (SBP), diastolic blood pressure (DBP), and maximum velocity of blood flow through the mitral valve (MV) were also quantitated. Flow-mediated dilation and echocardiography were measured using a SonoSite Edge II ultrasound (Fujifilm SonoSite, Bothell, WA, USA) with detection using a P10x transducer (8–4 Hz) to detect cardiac endpoints and the L38xi (10–5 Hz) transducer for vascular imaging. The following equation was used to calculate the flow-mediated dilation:%FMD = 100% × [(maximum diameter postcuff release) − (baseline diameter)]/(baseline diameter).

Two-dimensional guided M-mode echocardiography was used to obtain a right parasternal short-axis view of the heart at the level of the papillary muscles [66]. Measurements of left ventricular end diastolic inner diameter (LVID_d_) and left ventricular end-systolic inner diameter (LVID_s_) were also compiled from all dogs and normalized to body weight according to methodology described by Cornell et al. [66]. Finally, the N-Terminal pro–brain natriuretic peptide (NT-proBNP, Biosite, San Diego, CA, USA) and cardiac troponin (Canine hs-cTn1; Nordic Biosite, Täby, Sweden) levels were determined in plasma samples according to the manufacturer’s instructions for the respective kits.

### 4.6. Fecal Bile Acid Content

Bile acid levels in fecal samples were measured with Total Bile Acid Assay Kit (Cell-Biolabs Inc., San Diego, CA, USA, STA-631) according to the manufacturer’s instructions.

### 4.7. Statistical Analysis

Analyses were executed using SAS (version 9.4; SAS Institute, Cary, NC, USA). Prior to performing all analyses, the data were explored for normality and outliers using the PROC UNIVARIATE model in SAS and the Shapiro–Wilk test. NT-proBNP and cardiac troponin I data followed a non-parametric distribution and data was log-transformed to reach normality. One-way ANOVA was used to contrast differences among normal vs. high protein commercial diets and two-way ANOVA was used to contrast parameters for fava bean–based diets (fixed effects being fava bean variety and fermentation). In this current study, we used mixed-sex dogs to control the variation possibly connected with this factor. Preceding studies from our group have not discovered sex-related differences among (spayed/neutered) dogs [63] and because sex-related differences are not associated to the rationale of the present study, we did not incorporate this factor in the model. All post-hoc analyses were executed using the Fisher least significant difference (LSD) method. Differences were considered significant at *p* < 0.10.

## 5. Conclusions and Implications

One clear finding of the present study was that beagles fed high protein commercial, or moderate protein, fava bean-based diets did not show any tendency toward developing DCM. Moreover, fava bean-based diets did not cause hemolytic anemia. These results agree with our short-term trial in beagles. The period of one month does not provide enough time to draw conclusions related to long-term health. A period of 26 weeks is required to substantiate an adult maintenance claim for a dog food according to AAFCO regulations [67]. However, combined with the short-term study previously performed by the authors [19], these studies provide initial evidence to suggest FB could be included in dog food. However, feeding fava bean-based diets for the current one month period did adversely alter glucose tolerance in a variety-dependent manner, but this was counteracted by fermentation. Specifically, fermentation with *C. utilis* effectively reduced anti-nutritional factors and improved overall health of dogs mainly through increased energy and nutrient digestibility as well as increased RBC levels. The previously reported results showing that a high-protein commercial diet impaired blood chemistry results compared with the normal protein were not sustained in the present study. Moreover, dogs fed the high protein diet (a grain-free diet) showed no evidence of adverse cardiovascular functioning compared to a grain-containing normal protein diet, but this may be related to the added taurine in the high protein diet.

## Figures and Tables

**Figure 1 metabolites-11-00878-f001:**
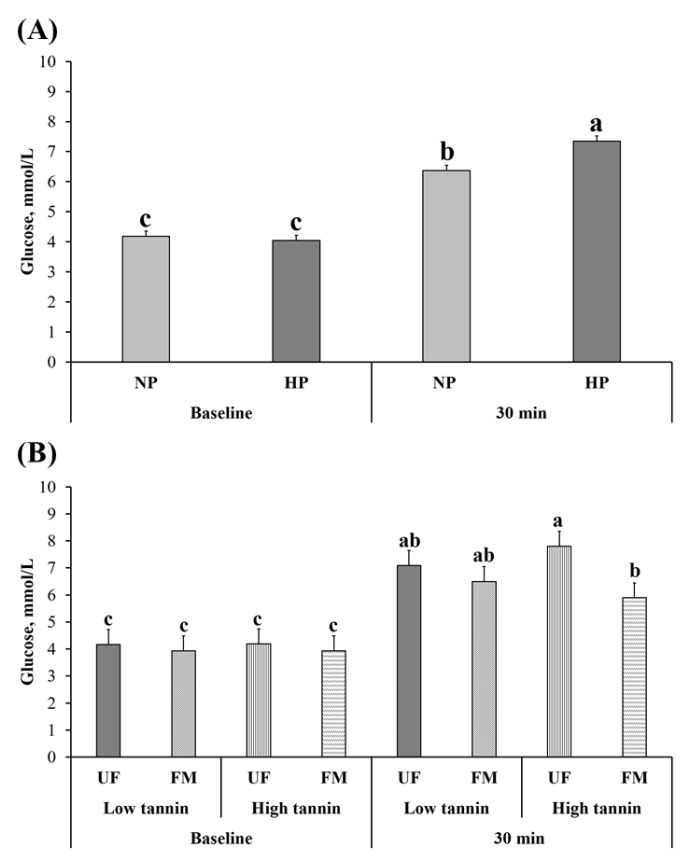
Oral glucose tolerance test in dogs fed either normal (NP) vs. high protein (HP) commercial diets (**A**), or low vs. high tannin fava beans-based diets without (UF) or with (FM) fermentation (**B**). Eight mixed-gender, neutered beagles were fed the NP or HP diets during the first and sixth months, respectively. From months 2 to 5, using a randomized, crossover, 2 × 2 Latin square design, 4 diets differing in fava bean variety and fermentation were compared as follows: low tannin-UF, low tannin-FM, high tannin-UF, high tannin-FM. Glucose levels correspond to baseline (0 min) and 30 min post-feeding glucose solution. a–c: Bars with different lowercase letters differ (*p* ≤ 0.05).

**Figure 2 metabolites-11-00878-f002:**
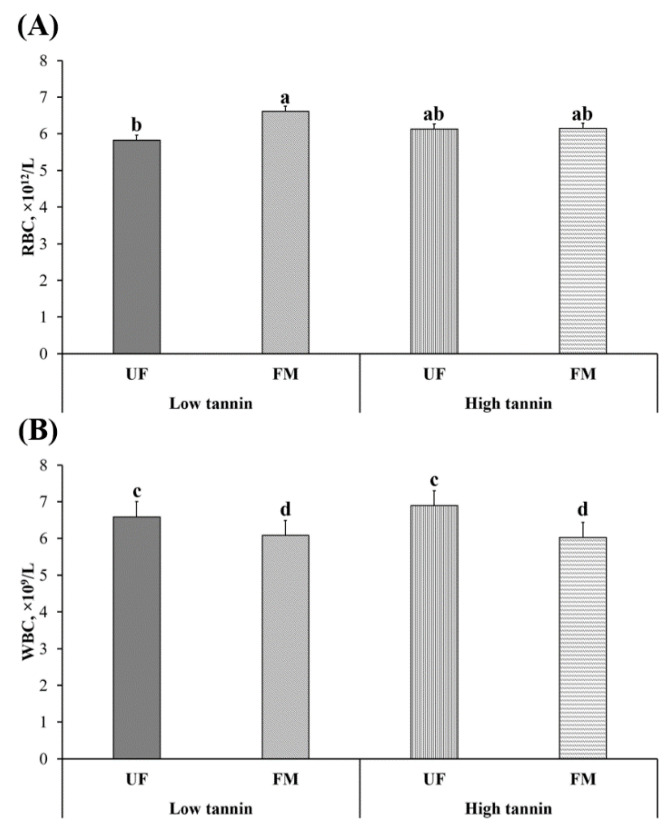
Red (RBC; **A**) and white (WBC; **B**) blood cell counts in dogs fed low vs. high tannin fava beans-based diets without (UF) or with (FM) fermentation. From months 2 to 5, using a randomized, crossover, 2 × 2 Latin square design, eight mixed-gender, neutered beagles were fed 4 diets differing in fava bean variety and fermentation as follows: low tannin-UF, low tannin-FM, high tannin-UF, high tannin-FM. a,b: Bars with different lowercase letters (*p* ≤ 0.05). c,d: Bars with different lowercase letters (*p* ≤ 0.10).

**Figure 3 metabolites-11-00878-f003:**
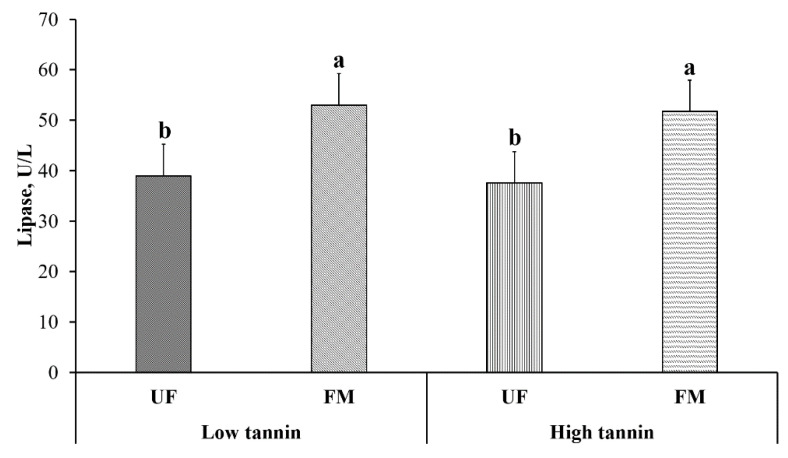
Blood lipase content in dogs fed either low or high tannin fava bean-based diets without (UF) or with (FM) fermentation. From months 2 to 5, using a randomized, crossover, 2 × 2 Latin square design, eight mixed-gender, neutered beagles were fed 4 diets differing in fava bean variety and fermentation as follows: low tannin-UF, low tannin-FM, high tannin-UF, high tannin-FM. a,b: Bars with different lowercase letters differ (*p* ≤ 0.05).

**Figure 4 metabolites-11-00878-f004:**
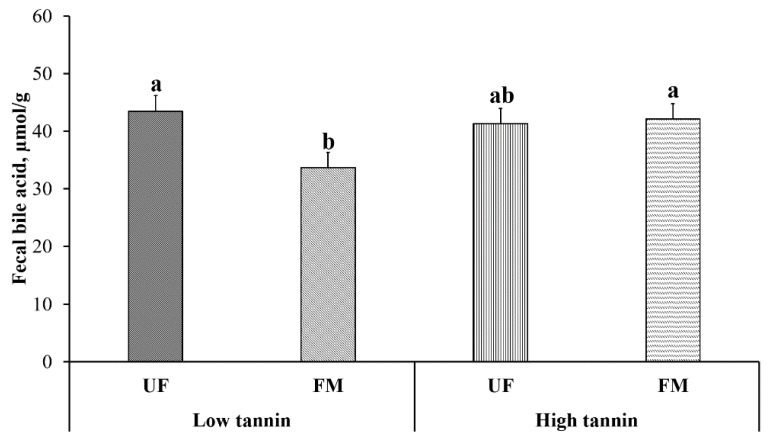
Fecal bile acid content in dogs fed either low or high tannin fava bean-based diets without (UF) or with (FM) fermentation. From months 2 to 5, using a randomized, crossover, 2 × 2 Latin square design, eight mixed-gender, neutered beagles were fed 4 diets differing in fava bean variety and fermentation as follows: low tannin-UF, low tannin-FM, high tannin-UF, high tannin-FM. a,b: Bars with different lowercase letters differ (*p* ≤ 0.05).

**Table 1 metabolites-11-00878-t001:** Body weight, food portion, and body condition score (BCS) of dogs fed diets formulated with either low or high tannin fava beans without (UF) or with (FM) fermentation, or normal (NP) vs. high (HP) protein commercial diets for one month each.

Item	Commercial			Low Tannin		High Tannin			*p* Value		
	NP	HP	SEM	UF	FM	UF	FM	SEM	P	FB	FM
Body weight (kg)	9.40	9.20	0.881	9.20	9.19	9.15	9.28	0.843	0.87	0.98	0.94
Food portion (g/d)	89.75	84.75	8.162	79.57	99.50	96.43	81.38	11.008	0.67	0.96	0.83
BCS	4.50	4.75	0.222	4.88	4.63	4.63	4.88	0.263	0.43	1.00	1.00

Eight mixed-gender, neutered beagles were fed the NP or HP diets during the first and sixth months, respectively. From months 2 to 5, using a randomized, crossover, 2 × 2 Latin square design, 4 diets differing in fava bean variety and fermentation were compared as follows: unfermented (UF) high tannin, fermented (FM) high tannin, UF low tannin, and FM low tannin. Values expressed as means (*n* = 8). SEM = pooled standard error of the mean. P = *p*-value between commercial diets with different protein content (NP vs. HP). FB = fava bean (low tannin vs. high tannin), FM = fermentation (UF vs. FM). The interaction between FB and FM was not significant for any of the parameters measured (*p* > 0.10).

**Table 2 metabolites-11-00878-t002:** Blood indicators of hepatic function in dogs fed diets formulated with either low or high tannin fava beans without (UF) or with (FM) fermentation, or normal (NP) vs. high (HP) protein commercial diets for one month each.

Item	Reference	Commercial			Low Tannin		High Tannin			*p* Value			
		NP	HP	SEM	UF	FM	UF	FM	SEM	P	FB	FM	FB*FM
Cholesterol (mmol/L)	2.70–5.94	4.98	4.54	0.304	3.37	3.72	3.44	3.65	0.269	0.32	0.99	0.29	0.80
TB (μmol/L)	1.0–4.0	1.14	1.05	0.111	1.61	1.35	1.64	1.89	0.355	0.58	0.42	0.97	0.47
DB (μmol/L)	0.0–2.0	0.46	0.58	0.055	0.61 ^a,b^	0.63 ^a,b^	1.01 ^a^	0.51 ^b^	0.036	0.17	0.85	0.11	0.04
IB (μmol/L)	0.0–2.5	0.77	0.48	0.075	0.66	0.73	1.06	1.01	0.194	0.02	0.07	0.96	0.77
ALP (U/L)	9–90	37.63	30.57	4.865	76.63	65.50	64.75	52.86	10.732	0.34	0.24	0.26	0.97
GGT (U/L)	0–8	1.25	1.63	0.457	2.75	2.14	2.85	1.57	0.540	0.57	0.66	0.08	0.52
ALT (U/L)	19–59	20.00	19.50	1.336	20.43	25.38	24.50	26.25	1.969	0.80	0.23	0.09	0.43
GLDH (U/L)	0–7	2.43	3.38	0.357	3.14	2.63	3.13	2.86	0.301	0.07	0.71	0.19	0.67
CK (U/L)	51–418	152.75	146.43	20.526	142.86	182.25	160.12	171.71	29.031	0.82	0.90	0.37	0.63
Total protein (g/L)	55–71	50.63	47.57	4.908	54.50	55.13	54.38	54.63	1.334	0.65	0.81	0.74	0.88
Albumin (A; g/L)	32–42	32.00	34.50	1.395	33.25	34.00	33.75	33.63	1.167	0.23	0.95	0.79	0.71
Globulin (G; g/L)	20–34	18.63	18.63	0.718	21.25	21.13	20.38	21.00	0.774	0.99	0.52	0.74	0.63
A: G	1.06–1.82	1.74	1.88	0.116	1.59	1.63	1.65	1.61	0.088	0.42	0.78	0.97	0.64
Urea (mmol/L)	3.5–11.4	5.56	7.26	0.251	5.88	5.83	6.14	5.61	0.365	0.02	0.36	0.78	0.66
Creatinine (mmol/L)	41.121	60.00	67.38	5.031	61.75	55.29	64.13	62.38	6.033	0.65	0.36	0.22	0.19

Eight mixed-gender, neutered beagles were fed the NP or HP diets during the first and sixth months, respectively. From months 2 to 5, using a randomized, crossover, 2 × 2 Latin square design, 4 diets differing in fava bean variety and fermentation were compared as follows: unfermented (UF) high tannin, fermented (FM) high tannin, UF low tannin, and FM low tannin. Values expressed as means (*n* = 8). SEM = pooled standard error of the mean. TB = total bilirubin, DB = direct bilirubin, IB = indirect bilirubin, ALP = alkaline phosphatase, GGT = gamma-glutamyl transferase, ALT = alanine aminotransferase, GLDH = glutamate dehydrogenase, CK = creatine kinase. P = *p*-value between commercial diets with different protein content (NP vs. HP), FB = fava bean (low tannin vs. high tannin), FM = fermentation (UF vs. FM), FB*FM = interaction between FB and FM factors. ^a,b^ Means within the same row with different superscript letters differ significantly (*p* < 0.05).

**Table 3 metabolites-11-00878-t003:** Blood electrolytes (mmol/L) of dogs fed diets formulated with either low or high tannin fava beans without (UF) or with (FM) fermentation, or normal (NP) vs. high (HP) protein commercial diets for one month each.

Item	Reference	Commercial			Low Tannin		High Tannin			*p* Value			
		NP	HP	SEM	UF	FM	UF	FM	SEM	P	FB	FM	FB*FM
Na	140–153	145.87	147.63	0.49	146.87	146.50	146.63	146.86	0.37	0.02	0.88	0.85	0.42
K	3.8–5.6	4.65	4.71	0.06	4.63	4.61	4.54	4.64	0.06	0.44	0.66	0.53	0.42
Na:K	28–38	31.25	31.38	0.37	31.63	31.75	32.25	31.75	0.41	0.81	0.45	0.65	0.45
Cl	105–120	114.00	111.38	0.42	109.75	94.43	109.50	109.25	6.73	<0.01	0.30	0.27	0.28
HCO_3_^−^	15–25	18.50	18.75	0.48	19.38	18.38	17.29	19.13	1.05	0.71	0.53	0.69	0.19
Anion gap	12–26	18.38	22.50	0.57	22.63	24.00	19.86	22.88	1.27	<0.01	0.14	0.09	0.53
Ca	1.91–3.03	2.42	2.90	0.35	2.48	2.49	2.46	2.49	0.03	0.33	0.77	0.38	0.84
P	0.63–2.41	1.14	1.24	0.04	1.43 ^a^	1.28 ^a,b^	1.23 ^b^	1.29 ^a,b^	0.05	0.08	0.08	0.43	0.05
Mg	0.70–1.16	0.74	0.80	0.02	1.14	1.45	1.00	1.28	0.41	0.01	0.70	0.48	0.97

Eight mixed-gender, neutered beagles were fed the NP or HP diets during the first and sixth months, respectively. From months 2 to 5, using a randomized, crossover, 2 × 2 Latin square design, 4 diets differing in fava bean variety and fermentation were compared as follows: unfermented (UF) high tannin, fermented (FM) high tannin, UF low tannin, and FM low tannin. Values expressed as means (n = 8). SEM = Standard error of the mean. P = *p*-value between commercial diets with different protein content (NP vs. HP), FB = fava bean (low tannin vs. high tannin), FM = fermentation (UF vs. FM), FB*FM = interaction between FB and FM factors. ^a,b^ Means within the same row with different superscript letters differ significantly (*p* < 0.05).

**Table 4 metabolites-11-00878-t004:** Cardiovascular function parameters of dogs fed diets formulated with either low or high tannin fava beans without (UF) or with (FM) fermentation, or normal (NP) vs. high (HP) protein commercial diets for one month each.

Item	Commercial			Low Tannin		High Tannin			*p* Value			
	NP	HP	SEM	UF	FM	UF	FM	SEM	P	FB	FM	FB*FM
EDV M-mode (mL/kg BW)	1.72	2.39	0.246	1.99	1.98	1.21	1.95	0.25	0.09	0.72	0.59	0.62
EDV Simpson’s rule (mL/kg BW)	1.44	2.11	0.144	2.08	2.08	2.11	1.92	0.104	<0.01	0.52	0.39	0.32
ESV M-mode (mL/kg BW)	0.55	0.45	0.062	0.45	0.57	0.46	0.66	0.104	0.25	0.91	0.08	0.92
ESV Simpson’s rule (mL/kg BW)	0.57	0.67	0.056	0.56	0.62	0.79	0.56	0.108	20	0.42	0.42	0.17
LVID_d_ (cm/BW^1/3^)	1.17	1.24	0.044	1.17	1.15	1.14	1.16	0.044	0.27	0.82	0.95	0.67
LVID_s_ (cm/BW^1/3^)	0.82	0.71	0.039	0.66	0.63	0.63	0.68	0.036	0.05	0.7	0.75	0.24
SV M-mode (mL/kg BW)	1.45	1.83	0.253	1.67	1.57	1.63	1.92	0.316	0.3	0.61	0.75	0.52
SV Simpson’s mode (mL/kg BW)	0.98	1.45	0.114	1.51	1.38	1.49	1.37	0.106	0.02	0.82	0.22	0.95
CO M-mode (L/min BW)	0.15	0.17	0.023	0.71	0.67	0.14	0.13	0.376	0.51	0.14	0.93	0.97
CO Simpson’s rule (L/min BW)	0.09	0.14	0.013	0.15	0.13	0.13	0.11	0.015	0.02	0.38	0.28	0.95
EF M-mode (%)	56.07	75.75	6.147	76.25	77	77.81	73.94	2.934	0.04	0.8	0.59	0.43
EF Simpson’s rule (%)	62.36	68.31	2.357	72.79	68	73.63	71	1.989	0.08	0.35	0.08	0.59
HR (bpm)	83.43	94.63	11.977	95.5	88.13	85	90.13	6.809	0.51	0.54	0.87	0.37
DWT (cm)	0.82	0.82	0.044	0.88	0.86	2.37	1.3	0.326	0.79	0.01	0.14	0.15
SWT (cm)	1.2	2.22	0.053	1.31 ^b^	1.33 ^b^	2.53 ^a^	1.34 ^b^	0.224	0.81	0.02	0.02	0.02
E/A ratio	0.18	0.19	0.014	0.15	0.16	0.18	0.16	0.242	0.63	0.69	0.68	0.65
MV (E wave) (cm/s)	71.61	69.31	12.067	80.53	69.16	67.93	77.2	10.55	0.9	0.83	0.92	0.33
MV (A wave) (cm/s)	23.15	37.03	4.306	40.39	36.46	31.49	29.59	3.651	0.05	0.04	0.45	0.79
SBP (mmHg)	136.4	127.25	7.128	117.19 ^b^	152.79 ^a^	132.31 ^a,b^	138.19 ^a,b^	10	0.38	0.97	0.16	0.05
DBP (mmHg)	75.73	60.5	3.948	67.19	81.96	66.31	75.58	6.016	0.01	0.55	0.05	0.65
FMD (%)	9.84	7.47	2.955	6.18	4.24	7.25	3.75	0.672	0.6	0.85	0.04	0.63
NT-ProBNP (pg/mL)	15.08	8.84	8.276	23.93	15.38	7.44	17.87	9.065	0.6	0.45	0.92	0.31
Cardiac troponin I (pg/mL)	5.23	2.98	2.655	8.14	4.94	7.86	17.11	9.352	0.55	0.53	0.75	0.51

Eight mixed-gender, neutered beagles were fed the NP or HP diets during the first and sixth months, respectively. From months 2 to 5, using a randomized, crossover, 2 × 2 Latin square design, 4 diets differing in fava bean variety and fermentation were compared as follows: unfermented (UF) high tannin, fermented (FM) high tannin, UF low tannin, and FM low tannin. Values expressed as means (*n* = 8). SEM = pooled standard error of the mean. EDV = left ventricular end-diastolic volume, ESV = left ventricular end-systolic volume; LVID_d_ = left ventricular end diastolic diameter, LVID_s_ = left ventricular end systolic diameter, SV = stroke volume, HR = heart rate, CO = cardiac output, EF = ejection fraction, DWT = left ventricle diastolic wall thickness, SWT = left ventricle systolic wall thickness, SBP = systolic blood pressure, DBP = diastolic blood pressure, FMD = flow mediated dilation, E:A ratio: ratio between E and A wave, MV (E wave) = maximum velocity through mitral valve (E wave), MV (A wave) = maximum velocity through mitral valve (A wave), NT-ProBNP = N-terminal pro brain natriuretic peptide. P = *p*-value between commercial diets with different protein content (NP vs. HP), FB = fava bean (low tannin vs. high tannin), FM = fermentation (UF vs. FM), FB*FM = interaction between FB and FM factors. ^a,b^ Means within the same row with different superscript letters differ significantly (*p* < 0.05).

**Table 5 metabolites-11-00878-t005:** Apparent total tract digestibility of dogs fed diets formulated with either low or high tannin fava beans without (UF) or with (FM) fermentation for one month each.

Item	Low Tannin		High Tannin			*p* Value		
	UF	FM	UF	FM	SEM	FB	FM	FB*FM
Crude protein	85.38 ^a,b^	86.63 ^a^	85.41 ^a,b^	84.20 ^b^	0.428	0.07	0.98	0.04
Fat	76.79	75.94	88.12	83.94	8.346	0.24	0.75	0.83
Non-fiber carbohydrates	96.23	97.05	96.46	96.74	0.298	0.88	0.07	0.36
Gross energy	91.10 ^b^	92.07 ^a^	91.37 ^a,b^	90.88 ^b^	0.321	0.15	0.46	0.03
Methionine	71.13	70.55	62.01	62.67	11.084	0.45	0.99	0.95
Cysteine	82.16	82.92	96.41	82.84	11.224	0.52	0.56	0.52
Taurine	81.32	91.85	91.00	90.35	6.514	0.51	0.43	0.37

From months 2 to 5, using a randomized, crossover, 2 × 2 Latin square design, eight mixed-gender, neutered beagles were fed 4 diets differing in fava bean variety and fermentation as follows: unfermented (UF) high tannin, fermented (FM) high tannin, UF low tannin, and FM low tannin. Values expressed as means (*n* = 8). SEM = pooled standard error of the mean. FB = fava bean (low tannin vs. high tannin), FM = fermentation (UF vs. FM), FB*FM = interaction between FB and FM factors. ^a,b^ Means within the same row with different superscript letters differ significantly (*p* < 0.05).

**Table 6 metabolites-11-00878-t006:** Plasma amino acid levels (nmol/mL) of dogs fed diets formulated with either low or high tannin fava beans without (UF) or with (FM) fermentation, or normal (NP) vs. high (HP) protein commercial diets for one month each.

Item	Commercial			Low Tannin		High Tannin			*p* Value		
	NP	HP	SEM	UF	FM	UF	FM	SEM	P	FB	FM
Taurine	105.38	91.25	10.712	84.71	73.57	77.43	78.75	12.652	0.36	0.93	0.70
Cystine	16.29	22.13	2.287	20.00	17.50	19.00	15.13	1.502	0.09	0.27	0.04
Methionine	43.00	41.43	6.629	46.25	53.71	55.00	46.62	4.864	0.86	0.86	0.92

Eight mixed-gender, neutered beagles were fed the NP or HP diets during the first and sixth months, respectively. From months 2 to 5, using a randomized, crossover, 2 × 2 Latin square design, 4 diets differing in fava bean variety and fermentation were compared as follows: unfermented (UF) high tannin, fermented (FM) high tannin, UF low tannin, and FM low tannin. Values expressed as means (*n* = 8). SEM = pooled standard error of the mean. P = *p*-value between commercial diets with different protein content (NP vs. HP), FB = fava bean (low tannin vs. high tannin), FM = fermentation (UF vs. FM). The interaction between FB and FM was not significant for any of the parameters measured.

## Data Availability

The data presented in this study are available on request from the corresponding author. The data are not publicly available due to ethical.

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
