# Peer review of "The Effects of Fermentation of Low or High Tannin Fava Bean-Based Diets on Glucose Response, Cardiovascular Function, and Fecal Bile Acid Excretion during a 28-Day Feeding Period in Dogs: Comparison with Commercial Diets with Normal vs. High Protein"

_metabolites, 2021, doi:10.3390/metabo11120878_

Round 1

Reviewer 1 Report

The paper entitled “The Effects of Fermentation of Low or High Tannin Fava Bean based Diets on Glucose Response, Cardiovascular Function, and Fecal Bile Acid Excretion during a 28-day Feeding Period in Dogs: Comparison with Commercial Diets with Normal vs. High Protein” evaluates the effects of treated faba beans based diets on several parameters in beagle dogs. The paper investigated an important issue of animal nutrition, especially in monostomached species, the possible use of vegetable protein sources without the negative effects of some anti-nutritional compounds. The paper is original and well structured. I suggest only minor revisions in order to improve its quality.
In particular:
The abstract could be rewritten, in order to focus on the obtained results of the study.
The introduction seems too long, I suggest to remodulate lines 49-79 and reduce it.
Why did you  choose these dosages of faba beans in the diets? This information should be given.
Table 2: I suggest to add a column with the physiological ranges for canine species of the analyzed parameters.

Reviewer 2 Report

The overall study design is acceptable with the exception of some oversight of adequate sample processing for cysteine analysis. My main issues are with the presentation of results and the conclusions as these appear to vastly exceed reasonable interpretation.

Specifically, the results do not refute the association between grain-free diets and diet related cardiomyopathy for several reasons:

  1. The length of the study- although this is a longer term study, it is still a feeding period that is too short to conclude that these diets are safe when fed for longer. It is possible that cardiac disease does not develop until a longer feeding period.
  2. This is a single experimental diet and the results cannot be extrapolated to every grain-free diet. We do not know what is the nature of the connection between diet and cardiomyopathy, and having a diet formulation that does not seem to lead to this, does not exclude a possible connection existing with other diets
  3. A single dog breed was evaluated
  4. No control group with no diet change to assess whether some reported echocardiographic changes can occur with time.

Additional comments:

  1. Abstract: the % of protein should be listed as a dry matter analyte
  2. Line 101: BCS is on a scale of 1-9? Should be cited as a scale by Laflamme et al.
  3. Figure 3: lipase is not sensitive or specific; therefore, unsure why the authors show lipase levels in a figure and how this would help the reader.
  4.   Line 186: Who performed the echocardiogram and what is their training? ‘One person’ does not tell the reader whether this was done by a boarded veterinary cardiologist with expertise in performing echocardiogram and interpreting the results.
  5. Lack of measuring cysteine in the plasma in this study is a strong limitation. Cysteine may be stabilized with sulfosalicylic acid; therefore the explanation regarding the lack of measurement should be that the samples were not treated as needed rather than ‘lack of stability’.
  6. Line 237 and elsewhere: ‘tended’ should not be used. Results are either significant or they are not.
  7. Line 271 and elsewhere: Since these are healthy dogs, are these changes in echocardiogram measures truly imply an improvement in cardiac function? Was this the interpretation of a veterinary cardiologist? Would these changes occur spontaneously in the same dog over time regardless of diet change?
  8. Line 296: the connection between lipase and fecal bile acids is conjecture and seems weak considering the lack of specificity of this parameter in dogs.
  9. Conclusions: See comments above regarding the refuting the connection between grain-free diets and DCM. Also- claiming safety after a short-term study is not acceptable. At minimum AAFCO feeding protocols last at least six months.

Reviewer 3 Report

Thank you to the authors for their excellent work. Studies such as this are necessary to continue the investigation into nutritionally-associated cardiac dysfunction in dogs. I have some general and some more specific comments, predominantly focusing on the cardiac investigations, below. 

Table 3 appears to be missing (is mentioned on Line 163). Without the table (and therefore values/reference values) it is not clear if the differences were clinically relevant (within or outside of the reference intervals) 

Line 183: “decreasing diastolic BP” is already stated above, line 182 

Line 188 - please define FMD (is only defined in table 4 caption and below in M&M as far as I can see) 

Table 5 (Line 236) - as you don't compare the commercial diets (NP vs HP) in this table, remove definition of "P" from the caption 

Line 325: "...indicates altered fermentability between Snowdrop and Florent" - Is this statement meant to indicate that there is altered fermentability between low- and high-tannin fava bean varieties? It could be stated more clearly. Also consider clarifying here that Snowdrop and Florent are 2 varieties of fava beans. Although you specify it below, it should be made clear when they are first mentioned.  

Line 325-326: "This corroborates previous findings from our group and others of an accelerated production of blood cells and increased health." - This sentence could be made more clear. Which finding results in accelerated blood cell production? Fermentation? If so please specify. Also specify "red" blood cells, as your results showed decreased white blood cells with fermentation. Finally, "increased health" is an extremely broad statement - are you implying that accelerated production of (red) blood cells results in increased health? In what way? Can this statement be backed up?  

Lines 408-409: Is it a benefit to have a higher A wave velocity? Why? This parameter in isolation doesn’t tell you much, and it is erroneous to infer that HP diets are beneficial to cardiac health because of increased A wave velocity.    

Lines 409-411: “In humans, increased left ventricular end systolic diameter has been associated with mitral valve dysfunction [47] which could lead to diastolic hypertension (increased diastolic blood pressure)”  

- This sentence is factually incorrect – neither an increased LV end systolic diameter nor mitral valve dysfunction can lead to increased diastolic blood pressure.  

- Additionally, what is the relevance of stating that LVESD has been associated with mitral valve dysfunction? Reference 47 states that in patients with organic mitral regurgitation (structural valvular disease) that an increased LVESD is associated with reduced survival. This is irrelevant to  your study as you are not reporting on dogs with structural valvular disease.  

Lines 432-433: It is likely more accurate to say that taurine supplementation is routinely used by veterinarians to improve cardiovascular function in patients with known or suspected taurine deficiency.  

Line 435: remove “the” in “...with the that...” 

Line 632 (and throughout): Did you measure NT-proBNP or pro-BNP? I believe these are not interchangeable, with proBNP being a precursor which is cleaved into NT-proBNP and BNP (the c-terminal). Please confirm which was measured and correct throughout.  

Line 661: As noted above, I would be careful about the claim that the high protein diet showed improvements in cardiovascular health. I do agree, however, with the first part of this sentence that there was no evidence of adverse CV function.  

Finally, as a general concern/comment: LV volumes (and therefore ejection fraction, stroke volume, cardiac output) calculated from M-mode is not considered an accurate method of estimating LV volumes. From the 2015 American Society of Echocardiography and the European Association of Cardiovascular Imaging guidelines (DOI: 10.1016/j.echo.2014.10.003): 

“Volume calculations derived from linear measurements may be inaccurate, because they rely on the assumption of a fixed geometric LV shape such as a prolate ellipsoid, which does not apply in a variety of cardiac pathologies. Accordingly, the Teichholz and Quinones methods for calculating LV volumes from LV linear dimensions are no longer recommended for clinical use.” 

The same recommendation of avoiding linear methods of calculating LV volume is made for dogs, with Simpson’s method of discs (SMOD) being the preferred technique. https://doi.org/10.1111/jvim.16089 

Ideally for your study, LV volumes and ejection fraction ideally should be measured and calculated using SMOD. If this is not able to be performed, at minimum a discussion of the limitations of volumes/EF from linear dimensions is required.  

If the LV volumes and EF are to be believed, it seems that the increased EF in the HP group is largely due to increased LV EDV (1.72mL/kg in NP vs 2.39mL/kg in HP, despite the difference being statistically insignificant) rather than relevant changes in LV ESV. This infers that there may not truly be an improvement in the dogs’ systolic function, but instead simply that there was an increase in diastolic volume (largely determined by preload, or circulating blood volume). This warrants some discussion, and with this in mind it may be a stretch to conclude that increased EF suggests improved cardiac health (line 423). 

Author Response

Thank you to the authors for their excellent work. Studies such as this are necessary to continue the investigation into nutritionally-associated cardiac dysfunction in dogs. I have some general and some more specific comments, predominantly focusing on the cardiac investigations, below.

Table 3 appears to be missing (is mentioned on Line 163). Without the table (and therefore values/reference values) it is not clear if the differences were clinically relevant (within or outside of the reference intervals)

This was an unfortunate omission on our part and we apologize. Table 3 added as requested (page 6).

Line 183: “decreasing diastolic BP” is already stated above, line 182

Correction made (line 204 in revised manuscript).

Line 188 - please define FMD (is only defined in table 4 caption and below in M&M as far as I can see)

Information added as requested (line 210 in revised manuscript).  

Table 5 (Line 236) - as you don't compare the commercial diets (NP vs HP) in this table, remove definition of "P" from the caption

Correction made as requested.

Line 325: "...indicates altered fermentability between Snowdrop and Florent" - Is this statement meant to indicate that there is altered fermentability between low- and high-tannin fava bean varieties? It could be stated more clearly. Also consider clarifying here that Snowdrop and Florent are 2 varieties of fava beans. Although you specify it below, it should be made clear when they are first mentioned. 

Information added and corrected as requested (lines 347-349 in revised manuscript).

Line 325-326: "This corroborates previous findings from our group and others of an accelerated production of blood cells and increased health." - This sentence could be made more clear. Which finding results in accelerated blood cell production? Fermentation? If so please specify. Also specify "red" blood cells, as your results showed decreased white blood cells with fermentation. Finally, "increased health" is an extremely broad statement - are you implying that accelerated production of (red) blood cells results in increased health? In what way? Can this statement be backed up?

Information included and clarified as requested (lines 350-352 in revised manuscript).

Lines 408-409: Is it a benefit to have a higher A wave velocity? Why? This parameter in isolation doesn’t tell you much, and it is erroneous to infer that HP diets are beneficial to cardiac health because of increased A wave velocity.   

We agree that the benefit of altered A-wave are unclear. The statements about benefits and discussion of the A-wave were removed from the Discussion (lines 433-434 in revised manuscript).

Lines 409-411: “In humans, increased left ventricular end systolic diameter has been associated with mitral valve dysfunction [47] which could lead to diastolic hypertension (increased diastolic blood pressure)” 

- This sentence is factually incorrect – neither an increased LV end systolic diameter nor mitral valve dysfunction can lead to increased diastolic blood pressure. 

- Additionally, what is the relevance of stating that LVESD has been associated with mitral valve dysfunction? Reference 47 states that in patients with organic mitral regurgitation (structural valvular disease) that an increased LVESD is associated with reduced survival. This is irrelevant to  your study as you are not reporting on dogs with structural valvular disease. 

Thank you for your comments and we agree that what was stated was incorrect. We have completely revised these sentences and replaced the two references in these sentences, with a stricter focus on dilated cardiomyopathy instead of bringing valve disease into the discussion (lines 434-438 in revised manuscript).

Lines 432-433: It is likely more accurate to say that taurine supplementation is routinely used by veterinarians to improve cardiovascular function in patients with known or suspected taurine deficiency.

Information added as requested (line 467 in revised manuscript).

Line 435: remove “the” in “...with the that...”

Correction made (line 468 in revised manuscript).

Line 632 (and throughout): Did you measure NT-proBNP or pro-BNP? I believe these are not interchangeable, with proBNP being a precursor which is cleaved into NT-proBNP and BNP (the c-terminal). Please confirm which was measured and correct throughout. 

We measured NT-proBNP. Corrections addressed in every mention throughout the manuscript.

Line 661: As noted above, I would be careful about the claim that the high protein diet showed improvements in cardiovascular health. I do agree, however, with the first part of this sentence that there was no evidence of adverse CV function. 

We agree and removed any mention of CV benefit, leaving the conclusion at no adverse effect of the HP grain free diet.

Finally, as a general concern/comment: LV volumes (and therefore ejection fraction, stroke volume, cardiac output) calculated from M-mode is not considered an accurate method of estimating LV volumes. From the 2015 American Society of Echocardiography and the European Association of Cardiovascular Imaging guidelines (DOI: 10.1016/j.echo.2014.10.003):

“Volume calculations derived from linear measurements may be inaccurate, because they rely on the assumption of a fixed geometric LV shape such as a prolate ellipsoid, which does not apply in a variety of cardiac pathologies. Accordingly, the Teichholz and Quinones methods for calculating LV volumes from LV linear dimensions are no longer recommended for clinical use.”

The same recommendation of avoiding linear methods of calculating LV volume is made for dogs, with Simpson’s method of discs (SMOD) being the preferred technique. https://doi.org/10.1111/jvim.16089

Ideally for your study, LV volumes and ejection fraction ideally should be measured and calculated using SMOD. If this is not able to be performed, at minimum a discussion of the limitations of volumes/EF from linear dimensions is required. 

We thank the reviewer for this comment. We had originally avoided making the cardiac function table (table 4) too big. However, in response to this reviewer’s request, we have added in our data from Simpson’s rule and volume calculations from 2/4-chamber biplane (TMOD). Most the results are in agreement with what was originally presented from the linear M-mode values, with a couple of very minor differences. See updated Table 4, Results (page 7). The overall conclusions about cardiovascular function have not changed, so edits to the Discussion were not necessary.

If the LV volumes and EF are to be believed, it seems that the increased EF in the HP group is largely due to increased LV EDV (1.72mL/kg in NP vs 2.39mL/kg in HP, despite the difference being statistically insignificant) rather than relevant changes in LV ESV. This infers that there may not truly be an improvement in the dogs’ systolic function, but instead simply that there was an increase in diastolic volume (largely determined by preload, or circulating blood volume). This warrants some discussion, and with this in mind it may be a stretch to conclude that increased EF suggests improved cardiac health (line 423).

We agree and have removed the references to improved cardiac function throughout the Discussion. We have also added a sentence that increased preload may have led to the increased LV end diastolic volume and diameter in the dogs on the HP diet, as suggested by this reviewer (lines 449-460).

Round 2

Reviewer 2 Report

The authors have sufficiently addressed most of my concerns; however, I am still not sure whether the cardiovascular changes should be interpreted as 'improvements' in health considering these are overall healthy dogs.  

Author Response

Reviewer #2

The authors have sufficiently addressed most of my concerns; however, I am still not sure whether the cardiovascular changes should be interpreted as 'improvements' in health considering these are overall healthy dogs. 

The sentences containing reference to improvement was changed in response to this review comment and to those of reviewer #3 (lines 28, 307, much of page 13 and 686).

Reviewer 3 Report

My previous comments have been addressed and the manuscript is improved. I have two very small additional comments (below). 

Line 346-346: Do you know that fermentation results in a higher production of RBCs? If there was higher #s of RBCs could it be that they have a longer lifespan or delayed destruction rather than increased production?

Line 686: add "diet" after "normal protein"?

Author Response

My previous comments have been addressed and the manuscript is improved. I have two very small additional comments (below). 

     We are happy this reviewer is satisfied with our changes.

Line 346-346: Do you know that fermentation results in a higher production of RBCs? If there was higher #s of RBCs could it be that they have a longer lifespan or delayed destruction rather than increased production?

     You are correct that we do not know whether it is RBC production or longevity that is changed. We have edits these lines accordingly.

Line 686: add "diet" after "normal protein"

     Inserted as requested.